# Preliminary Evaluation of a Mobile, Web-Based Coaching Tool to Improve Pre-K Classroom Practices and Enhance Learning

Caroline Christopher [1,*], Sandra Jo Wilson [2], Mary Wagner Fuhs [3], Carolyn Layzer [2] and Sophie Litschwartz [2]

[1] Department of Teaching and Learning, Vanderbilt University, Nashville, TN 37235, USA
[2] Abt Associates, Cambridge, MA 02138, USA; sandra_wilson@abtassoc.com (S.J.W.); carolyn_layzer@abtassoc.com (C.L.); sophielitschwartz@gmail.com (S.L.)
[3] Department of Psychology, University of Dayton, Dayton, OH 45469, USA; mfuhs1@udayton.edu
[*] Correspondence: caroline.h.christopher@vanderbilt.edu

**Abstract:** Educators rely on professional development to improve instruction. Research suggests that instructional coaching which utilizes specific coaching practices, such as classroom observation followed by debriefing and goal setting, and integrated strategies such as co-teaching, bring about significant change in instructional practices. The goal of this study was to gauge whether or not the use of a web-based data collection and coaching tool led to changes in focal classroom practices and whether or not improving those practices was, in turn, related to students' academic and self-regulation gains across the prekindergarten year. To examine the implementation and impact of the coaching app, researchers conducted a cluster-randomized trial, comparing the classroom practices of teachers receiving business-as-usual coaching to those being coached with the app. Classroom observation data showed no significant differences in teachers' practices across the school year, and student achievement did not differ between conditions. Qualitative data from coach interviews, however, revealed that coaches using the app were more likely to employ integrated coaching strategies associated with improving instruction. The lack of differences in terms of teachers' practices and students' assessment gains may be due to a lack of statistical power and inconsistent professional development implementation associated with ongoing disruptions due to the pandemic. Further research examining the effectiveness of educational technologies supporting professional development is needed.

**Keywords:** instructional coaching; job-embedded professional development; prekindergarten teachers; classroom practices; educational technology; data-driven coaching



## 1. Introduction

Children's early learning experiences have long-term effects on their academic and social–emotional development. Research investigating these effects has found that attending high-quality prekindergarten (Pre-K) programs is associated with lower grade retention and increased high school graduation rates [1], greater earnings, better health, and less involvement in crime in adulthood [2]. Importantly, these benefits are even more pronounced for students from economically disadvantaged backgrounds [3,4], suggesting that early learning experiences may help to close the opportunity gap that exists between lower and higher socioeconomic students. Moreover, cost–benefit analyses of high-quality Pre-K points to substantial economic benefits for society at large [5,6]. For these reasons, it is critical that educators provide high-quality learning opportunities in Pre-K, most children's first exposure to an organized learning environment, which can be built upon in subsequent grades. To accomplish this, educators need ongoing support as they are relied upon to set students on a positive trajectory.

In recent years, there has been a push to find effective approaches to professional development that are job-embedded and focus on iterative and experiential learning, including

strategies such as observing role models, co-planning lessons, reflecting on practices, and receiving continual feedback [7]. Instructional coaching, involving coaches' and teachers' co-constructed goal setting conversations, is one such approach that has become popular, in no small part due to the increasing evidence of its effectiveness for improving content-specific instruction as well as using evidence-based instructional and behavior management strategies [8]. For example, Crawford et al. [9] found that teachers receiving coaching showed improvements in strategies to support print and letter knowledge and writing compared to teachers receiving other forms of professional development. Another study found that the use of data-driven coaching that focused on social–emotional competencies and included the provision of performance feedback for teachers was associated with students' self-regulation gains [10]. Coaching focused on instructional strategies has been shown to support teachers' use of specific praise and academic performance feedback (e.g., teachers' ability to highlight key concepts that arose during instruction) [11]. Importantly, research on teachers' perceptions of coaching indicated that teachers' self-efficacy was higher among those who received coaching [12]. Walsh et al. [12] found that this was particularly true for teachers in the early grades. Moreover, teachers with three or fewer years of experience perceived that coaching made an overall greater impact on their instruction, helped improve their classroom management, helped them promote greater student engagement, and improved their instructional strategies.

While the proximal goal of any professional development activities is to increase knowledge and improve classroom practices, the ultimate goal is to promote student success. Instructional coaching is an effective approach that can achieve both. For example, a meta-analysis provided evidence that changes in classroom practices after teachers receive coaching are associated with students' academic achievement [13]. Additionally, while some studies do not find a direct effect of coaching on students' outcomes, there is evidence that changes in instruction could mediate the relationship between coaching and students' academic gains [9]. In addition, the literature suggests that coaching is beneficial amongst both general education and special education teachers [14], suggesting that it provides a framework that can be used flexibly to provide support across a range of contexts.

### 1.1. Effective Coaching

Researchers have investigated the mechanisms that drive success in coaching and found that coaching which utilizes specific productive coaching practices, such as coach planning, classroom observation followed by reflection and goal setting, and progress monitoring over time, brings about significant change in instructional practices [15]. In addition to these coaching behaviors, there is evidence that coaching that targets specific (and measurable) classroom practices, such as behavior management, is more effective than just the provision of other types of instrumental support (e.g., when a coach offers to work individually with a child while the teacher leads a whole group activity with the rest of the class) [16,17].

A coaching cycle that includes classroom observation followed by a debrief conversation between the coach and teacher and goal setting is considered the best practice. Researchers have identified characteristics of these coaching conversations affect the amount of growth teachers experience. Reflection has been found to be an active ingredient in effective coaching in numerous studies [18–20]. For example, Witherspoon et al. [17] compared coaching conversations of coach–teacher pairs in a high-growth group to those in a the low-growth group (defined based on measures of instructional quality across two years), and found that high-growth pairs had conversations in which the coach asked questions in a way that was more likely to lead to greater teacher reflection and overall participation and input in the conversation and goal setting.

### 1.2. Linking Classroom Quality to Child Outcomes

Observation data can be used to prompt teachers' reflection on their instructional practices which can then inform targeted goal setting. Snyder and Delgado [21] found that

for coaching to be successful (i.e., for it to improve instruction and student outcomes), it should be informed by data that are directly tied to goal setting. The importance of this is highlighted in recent research in which coaches were asked to rank their own ongoing professional development needs; coaches consistently reported that support for helping teachers develop targeted goals was one of their top needs [22]. Despite evidence of the importance of assessing teacher practices to inform coaching, relatively few coaching models consider the collection of systematic classroom observation data central to their process [23].

Moreover, when tools are used to systematically collect data, the most common measures of classroom quality rely on rating systems. In early childhood, some of the common observation rating systems include the Early Childhood Environmental Rating Scale (ECERS) [24] with several further editions and the Classroom Assessment Scoring System (CLASS) [25]. Recent summaries of the research on these two systems for rating quality concluded that neither system was strongly or consistently associated with growth in any of the various achievement areas [26]. Others have suggested that these instruments may simply not be strong enough as measures of quality [27] or that the problem lies in the difficulty of using rating-based systems reliably [28]. Ratings by definition have a subjective component; it can be extremely difficult for raters to maintain reliability in the field. Further supporting an argument in favor of objective measurement, recent research indicates that, when compared to rating scales, data collected using specific behavioral count measures provide stronger associations with child outcomes [29].

### 1.3. Leveraging Technology to Support Professional Development

In recent decades, the educational technology (ed tech) industry has advanced dramatically, and with good reason. There is substantial evidence that ed tech that is focused on promoting student achievement in a plethora of content areas can produce positive results [30], particularly when an instructor is trained to use the specific tool, and facilitates students' use of the educational technology. Moreover, there is some evidence that lower-achieving students may benefit from interactive and adaptive education technologies even more than higher-achieving students do [31].

Beyond students' direct interaction with instructional videos, websites, and web-based applications, educational technology shows promise for supporting teachers' professional development [32]. For example, Walker et al. [33] found that tech-based professional development to support teachers' strategies for designing online math and science learning activities for students was effective at improving teacher knowledge, skills, and technology integration. Similarly, Burstein et al. [34] found that an app focused on providing teachers with strategies for adapting their lesson plans to help English learners address text-based challenges in learning content from readings improved teachers' knowledge of language barriers and ability to develop higher-quality lesson plans. Another group found that a web-based professional development program enhanced early career teachers' positive attitudes toward incorporating tech resources in their pedagogy [35].

However, the vast majority of ed tech tools designed for professional development focus on directly interacting with the teacher, without providing a way to take into account that teacher's current practice, which (1) typically requires classroom observations, and (2) is an important context for informing coaches' tailored support for teachers. Given that the majority of classroom observation tools are not systematic or are based on rating scales that are vulnerable to observer drift, collecting concrete data that can be reliably used to track progress over time is a challenge.

Pre-K teachers need valid information about their classroom practices before they can take steps to improve these practices. This information should be easily tied to specific classroom practices in order to identify behaviors to target change. Instructional coaches are in the position to be able to provide this type of feedback, but in order to do so coaches need to know when to observe in a classroom, what to look for, a way to collect information (i.e., data) systematically on what they see, and an understanding of how to use the

information to inform their coaching of teachers. The present study includes findings from a preliminary evaluation of a web-based data collection and coaching tool designed for use in Pre-K classrooms.

### 1.4. Background

In 2014, researchers began a four-year research–practice partnership with three early learning centers serving 27 Pre-K classrooms, which focused on identifying classroom practices that were related to students' gains. The partnership identified and then confirmed eight clusters of classroom practices that were linked to children's gains across a number of different domains [36]. The classroom practices include the following:

1.  Reducing time in transitions and promoting effective use of time.
2.  Increasing the quality of instruction. High-quality instruction involves opportunities for students to reflect, predict and communicate understanding.
3.  Creating a more positive emotional climate in the classroom.
4.  Increasing teachers' listening to children during instructional activities.
5.  Facilitating children's sequential activities (i.e., activities with predictable steps).
6.  Fostering associative and cooperative interactions during center-based activities.
7.  Fostering higher levels of involvement from children.
8.  More time on early mathematics—particularly counting and cardinality, geometry, measurement and operations.
9.  Incorporating more literacy for children, with a focus on unconstrained skills including comprehension and foundations of reading, language and writing.

Having identified and replicated findings that underscore the importance of these practices, our partner schools adopted these as the focus of their professional development. *A ninth practice, incorporating more literacy for children, with a focus on unconstrained skills, was an added focus beyond the initial eight identified during the partnership. Researchers subsequently partnered with instructional leaders to create a professional development series for coaches to use with their Pre-K teachers to improve these practices. Coaches then used the series, which comprised brief presentations and accompanying print materials, in their work with teachers. We continued to collect data on classroom practices and found that although teachers improved in some areas (e.g., reducing time in transitions), other areas were more difficult (e.g., increasing the level of instruction by, for example, asking inferential questions). In debriefing with coaches regarding their work with teachers, they cited a key limitation. The classroom observation data informing their goal setting with teachers had been collected by researchers using a complicated protocol that coaches found difficult to describe to teachers. Coaches did not "own" the data. Moreover, when they received summaries of observation data, they found it difficult to know how to use results to set specific goals with their teachers.

To address this need, we partnered with teachers, coaches, administrators, and engineers to design and develop a new coaching tool. Coaching to Help Activate Learning for Kids (CHALK) [37] is a web-based real-time data collection and feedback application for preschool instructional coaches and teachers, created for use on tablets. It was designed through an iterative process of working with target end users from a variety of early learning settings to develop a tool that would be user-friendly and provide a structure for collecting meaningful classroom observation data with embedded guidance on linking results to actionable goals.

### 1.5. CHALK Tool

CHALK supports data-driven goal setting with a library of questions and prompts that help inform the direction of post-observation debrief meetings and action planning and promotes transparency in the process by allowing all users—coaches and teachers—to create their own accounts and access all embedded features (see Supplementary Materials). The CHALK tool was developed by an interdisciplinary team of education researchers, software developers, early childhood administrators, teachers, and instructional coaches.

First, the user logs into a website. From there, they have access to training materials to help them identify and collect data on specific classroom practices, presented above, identified as being predictive of students' gains. After watching instructional videos focused on the practices and 'how to' videos on using the tool to collect data on those practices, they 'unlock' the ability to collect observation data on a given practice. When an observation has been completed, the user sees instant graphical depictions of results and prompts to aid in their interpretation of the data. At this point, they can see and select from the library of coaching questions as they plan for their debrief meeting with the teacher. In the meeting, a combination of viewing the data together and using prompts helps encourage teachers to reflect on their practice. An embedded action planning tool with guidance on how to set goals and define action steps can be filled out by the coach and teacher together and then shared, ensuring that the coach and teacher have access to action plans through their respective CHALK accounts. The coach–teacher team can then gauge progress over time through continued data collection and viewing results in trends graphs.

*1.6. Current Study*

The goal of the study was to gauge whether or not the use of a web-based data collection and coaching tool led to changes in focal classroom practices and whether or not improving those practices was, in turn, related to greater academic and self-regulation gains across the Pre-K year. Thus, we collected quantitative and qualitative data to address the following research questions.

Research Questions: Implementation

1. Are there differences in the focus of coaching sessions, between CHALK and standard coaching conditions?
2. Are there differences in the coaching process overall, between CHALK and standard coaching conditions?

Research Questions: Impact

1. After one year of coaching, do Pre-K teachers with CHALK coaches exhibit greater improvements in the nine targeted classroom practice areas than Pre-K teachers who receive coaching-as-usual do?
2. After one year of exposure, do Pre-K students in the CHALK condition exhibit greater improvements in mathematics, literacy, language, and executive function than Pre-K students in the business-as-usual coaching condition do?

## 2. Materials and Methods

*2.1. Participants*

The analytic sample included four coaches and fifteen classrooms in the CHALK condition, and three coaches and ten classrooms in the control condition. One classroom in the control group was led by 2 co-teachers; thus, while there were thirty-five classrooms in the sample, the number of teachers was 36. Table 1 presents the teachers' demographic characteristics. All participating instructional coaches were female. Roughly, a third of the CHALK coaches were Black while two-thirds of the usual coaching group was Black. All 36 teachers were female and about half were Black. Teachers' degree/credential status ranged from those with Child Development Associate (CDA) certificates to those with master's degrees. The years of experience for teachers ranged from new teachers with 1–2 years of experience to those with more than 10 years of experience in the profession.

The sample also included 208 children, with 119 in the CHALK condition and 89 in the control group. Child gender was balanced across the CHALK and control samples, but there were substantial racial and age imbalances across conditions. The CHALK sample was 60% Black and 34% White, whereas the control sample was 80% Black and 17% White. The two conditions were also imbalanced in terms of child age. The CHALK students were two months older on September 1st of the school year and one month older at assessment than control students. Students' demographic characteristics are included in Table 2. Methods

of quantifying the magnitude of differences between the students in each condition are included in the results.

**Table 1.** Teacher background characteristics.

|  | CHALK | Business-As-Usual Coaching |
|---|---|---|
| Proportion of female teachers | 1.0 | 1.0 |
| Proportion of Black teachers | 0.33 | 0.63 |
| Proportion of White teachers | 0.58 | 0.26 |
| Proportion with 1–2 years of experience | 0.06 | 0.17 |
| Proportion with 3–5 years of experience | 0.17 | 0.17 |
| Proportion with 6–10 years of experience | 0.22 | 0.17 |
| Proportion with more than 10 years of experience | 0.56 | 0.50 |
| Proportion with a high school degree/GED | 0.12 | 0.20 |
| Proportion with an associate's degree | 0.25 | 0.20 |
| Proportion with a bachelor's degree | 0.21 | 0.40 |
| Proportion with a master's degree | 0.12 | 0.13 |
| Proportion with a CDA | 0.29 | 0.07 |

**Table 2.** Demographic characteristics of participating children.

|  | CHALK ($n$ = 119) | Business-As-Usual Coaching ($n$ = 89) | Overall | ES | *p*-Value |
|---|---|---|---|---|---|
| Proportion of female children | 0.53 | 0.54 | 0.53 | −0.01 | 0.96 |
| Proportion of Black children | 0.60 | 0.80 | 0.68 | −0.61 | 0.32 |
| Proportion of White children | 0.34 | 0.17 | 0.27 | 0.55 | 0.36 |
| Age in months on 1 September 2021 | 49 (7.1) | 47 (7.1) | 48 (7.2) | 0.32 | 0.25 |
| Age in months at baseline assessment | 52 (7.6) | 51 (7.0) | 52 (7.3) | 0.12 | 0.66 |
| Mean number of days between Sept 1 and pretest | 79 (29) | 100 (43) | 88 (37) | −0.60 | 0.17 |
| Mean number of days between pretest and posttest | 141 (28) | 118 (37) | 132 (34) | 0.73 | 0.08 |

Notes: Standard deviations in parentheses. Effect sizes for proportion of female children, proportion of Black children, and proportion of White children were estimated using the Cox transformation [38], and effect sizes for age in months on 1 September 2021, age in months at assessment, and mean number of days between 1 September 2021, and the assessment date were estimated using Hedges' *g* [39].

### 2.2. Procedure

Seven instructional coaches were recruited through a preschool provider in a medium-sized Midwestern city to participate in the experiment. After consenting to participate in the study, coaches were block-randomized to conditions based on the preschool centers they served, with four assigned to use CHALK and three in the business-as-usual condition. CHALK coaches were trained to use the real-time feedback application to work with their assigned classrooms as part of their regular job duties, and three coaches were assigned to continue their usual practices with their assigned classrooms. Participating coaches assisted in recruiting teachers from among those participating in their professional learning communities (PLCs).

All coaches participated in regular PLCs. Teachers were encouraged to enroll in a PLC that matched their interests and professional learning goals. The PLC activities were co-facilitated by (and often written by) the instructional coaches. Each PLC participant received a binder of materials, and some PLCs also provide kits for "make and take" activities to carry out at home. PLC meetings included 10–20 teachers and 2–3 co-facilitators and lasted from 90 min to 2 h, in the evenings. During the study year, PLC activities were conducted via Zoom, and included PowerPoint slides and videos as well as some more interactive formats in breakout rooms such as hands-on activities (e.g., creating materials, matching English language development standards with activities and an assigned technology), games, small group discussion, and whole group opportunities for presentation, questions and discussion.

Coaches in the CHALK condition were expected to fulfill all their usual responsibilities (lead a PLC, provide coaching linked to the PLC topics, observe their teachers in their classrooms, debrief, report to leadership on coaching activities, and support teachers in meeting the goals they laid out for themselves) in addition to using the tool. CHALK coaches received an initial training and a follow-up booster on the use and application of the CHALK tool; coaches were encouraged to communicate with the developer's team with questions, suggestions for improvement, and general comments. In addition to their usual responsibilities, CHALK coaches were expected to identify teachers' instructional needs that fall within the classroom practices addressed by the tool, use the CHALK tool as part of their regular classroom observation, use results from the tool as part of the follow-up conversation with the teacher about the observation (debrief), and use that coaching conversation to co-construct an action plan with the teacher. They were not instructed to focus on specific practices or to adhere to a firm schedule as to the frequency of their CHALK observations, debrief conversations, or the number of embedded practices to explore.

### 2.3. Teacher Measures and Data Collection

Classroom Observations. Trained observers from the field-based data collection team collected all observation data digitally on iPads. Each teacher in the study sample was observed three times over the 2021–2022 school year—once in the fall, again in early spring, and a third time later in the spring semester. The Teacher Observation in Preschool (TOP) protocol was used to measure observable aspects of teachers' classroom behaviors. The TOP protocol is completed in tandem with the *Child Observation in Preschool* (COP) protocol [40,41], which is used to measure observable child behaviors. The TOP protocol is collected via a series of snapshots of teacher behavior across the school day when children are in the room. The teacher's behavior is observed within a 3 s window and then scored on a series of dimensions. Once scoring has been completed for the teacher, the same procedure is followed for any assistants or other adults in the classroom. Children are coded in the same way immediately after. The TOP protocol measures how much and to whom the teacher talks and listens, the types of tasks (e.g., instruction, management, behavior approving or disapproving) in which the teacher is engaged, the level of instruction (none, low, basic skills, partly inferential, highly inferential), the areas of learning on which the teacher focuses (e.g., math, literacy, art, drama, or none), and the tone of the interactions the teacher has with the class.

The COP protocol, collected in tandem with the TOP protocol, is a system for observing children's behaviors in preschool classrooms across a daylong visit. A specific child is observed within a 3 s window and then coded across 9 dimensions; the observer then moves on to the next child. In an observation session, observers will sweep over all adults and children in the classroom up to 20 times. The COP measures how much and to whom the children talk and listen, the learning settings in which the children are observed (whole group, etc.), the different types of learning foci of the activities, and the level of involvement of the children.

Interviews. Each coach and teacher participated in a semi-structured interview in late spring. Coaches and teachers in both conditions had a common set of interview questions focused on how coaches and teachers selected topics for their coaching work and coaches were asked to report the types of coaching activities in which they engaged (coaching debrief conversations, strategies coaches used to support teachers' progress, etc.).

Prior to conducting the coach interviews, the study team reviewed the coach log data to identify the teachers with whom coaches were most actively engaged. The team then contacted the selected teachers via email to ask them to participate in interviews, on a voluntary basis. Thirteen teachers agreed to be interviewed and were interviewed. Interviews lasted between 45 min to 1 h.

Surveys. The study team emailed the participating coaches and invited them to complete a brief online survey. The surveys asked coaches to select coaching strategies

they regularly used with their teachers from a list of specific strategies including observing teachers, sharing resources, delivering professional development, modeling lessons, providing observation feedback, discussing classroom observation data, discussing student achievement, reviewing a classroom video, creating goals with teachers, and writing lesson plans.

Coaching logs. In order to obtain comparable reports of coaching interactions across conditions, we created a simple online reporting tool, the coach log, which coaches were asked to complete every week. Each coach received a unique link to their coach log that included a dropdown menu with the name of the teachers she was coaching who had consented to participate in the study. The log included the following fields: date of interaction, teacher with whom the coach interacted, mode of interaction ("What was the type of meeting or interaction?" with options such as in-person or teleconference), purpose of interaction (with options such as observation and post-observation debrief), and topics addressed.

### 2.4. Student Measures and Data Collection

Student Assessments. Each student in the study sample was assessed by a trained assessor at the beginning of Pre-K (pretest) and at the end of Pre-K (posttest) in the following domains: mathematics, executive function skills, literacy, and language.

Mathematics:

- The number sense subtest of Woodcock–Johnson III (WJ III) [42] assesses children's counting, problem solving, mathematical knowledge, and basic computation skills.
- The quantitative concepts subtest of WJ III requires pointing to or stating answers to questions on number identification, sequencing, shapes, symbols, terms, and formulas. It measures aspects of quantitative reasoning and math knowledge.

Executive Function:

- Head Toes Knees Shoulders (HTKS) [43] is a measure of inhibitory control, working memory, and attention focusing. Children are asked to play a game in which they do the opposite of what the assessor says to do (e.g., the assessor says to touch their toes, so the child must touch their head instead). McClelland et al. [44] examined the interrater reliability of the measure when used with Pre-K children and found it was high ($\kappa = 0.90$). We used the adapted revised version (HTKS-R) of this measure that provides a better floor for lower-performing children [44]. Using a sample of 169 preschoolers, McClelland et al. found that the revised measure is significantly correlated with other measures of self-regulation/executive function (WJ working memory: ($r = 0.19$ *) and day–night: ($r = 0.26$ **). It is also significantly correlated with academic outcomes (WJ letter–word: $r = 0.29$ **, applied problems: $r = 0.51$ ***, picture vocabulary: $r = 0.31$ ***).

Literacy:

- The WJ III letter–word identification subtest involves the identification of letters and reading of words. It requires identifying and pronouncing isolated letters and words.

Language:

- The WJ III picture vocabulary subtest is used to assess child expressive vocabulary. It is a standard measure that requires children to name pictures.
- The test–retest reliability of each of the WJ III subtests is greater than $r = 0.80$. Assessments were administered individually in a single session by trained and certified assessors who were blind to the experimental condition.

### 2.5. Analysis Plan

To address the research questions for the implementation study, we used a mixed-methods approach. We examined qualitative data from interviews in conjunction with descriptive statistics reported as available from surveys, coaching logs, and usage data from the CHALK tool itself. Usage data included the number of observations collected by

each coach and the classroom practice that was the focus of each observation (e.g., classroom climate).

We assessed the impact of CHALK by comparing the covariate-adjusted mean outcome values between the CHALK and usual coaching groups. We evaluated the impact of the intervention on the classroom observation measures by comparing mean values across the CHALK and usual coaching conditions at the intermediate and year-end time points.

The child assessment analysis used a complete case sample of children who had both pretest and posttest assessments. We used a two-level model with a random intercept clustered at the coach level (i.e., the level of the randomization). The model also contained covariates for the baseline assessment value, child age at the baseline assessment, days between the start of the school year and the baseline assessment, days between the baseline and endline assessment, child race, and child gender.

## 3. Results

We first compared the demographic characteristics of our participants at the baseline, calculating effect sizes between conditions using the COX transformation [38]. Child gender was balanced across the CHALK and usual coaching samples, but there were substantial racial and age imbalances across conditions. The CHALK sample was 60% Black and 34% White, whereas the usual coaching sample was 80% Black and 17% White. We then calculated effect sizes for age in months and age in months at assessment date and compared the number of days between pretest and posttest assessments by estimating Hedges' $g$ [39]. We found that the two conditions were also imbalanced in terms of child age. The CHALK students were two months older on September 1st of the school year and one month older at assessment than students in the usual coaching condition. This translates to a standardized mean difference of 0.32 between the child ages on September 1st and a standardized mean difference of 0.12 at assessment. This was due to the baseline child assessments being collected later, on average, for the usual coaching group than for the CHALK group. The mean assessment day for the CHALK group was 79 days after September 1st, and the mean assessment day for the usual coaching group was 100 days after September 1st ($g$ = 0.60). This produced average ages that were closer together at assessment for the two conditions but also meant that children in the usual coaching condition had more time in preschool before their baseline assessments were collected than did children in the CHALK condition. None of the demographic or assessment timing differences were statistically significant at the 0.05 level.

### 3.1. Implementation Study

Focus of coaching. All coaches and teachers reported that their primary focus for coaching was their respective PLC topic, but five of the coaches added that they used each observation and subsequent conversation as an opportunity to assess teachers' individual instructional needs and then adjusted the focus to fit teachers' needs. Six of the seven coaches described choosing a coaching focus in collaboration with their teachers (two CHALK coaches and all of the control group coaches).

> "So with the PLC that they're in, the topics are already laid out because we do a book study. It is—it's with what we're going through in the book. So that's how we do it. But if I was going in to observe the classroom, I would take that tool and say, okay, is she doing any number? And it's good because now I'm more aware and aware is the big thing. And intentional and being aware is the game in this tool". (chalk coach)

> "My PLC topic is very important and I feel like that I just can't get enough of it. And it's so good information that every time we review it, we get a new piece". (chalk teacher)

> "So I think it depends on their goals or what they're specifically working on. So who can I think of? Well, I can think of [Teacher], the girl who I talked about. She's been working on, with her large group time, student engagement". (chalk Coach)

Two coaches indicated that their PLC topics were designed to be applied to any or all areas of the classroom and instruction. Four of the seven coaches said they came to the classroom with a plan but could shift their plan depending on the classroom and/or the teacher's interests/needs, including the teacher's own objectives. They would base the choice of focus on an observation-based needs assessment, discussion with the teacher, and in one instance, input from a building administrator.

> "*So with the Purposefully Planning PLC, it's a bit easier because we are very—it's a little bit of room to wiggle, but it's very much focused. So we have the homework and the tasks to focus . . . So usually, I go and observe the classroom and see—and I put the lens of the homework or the focus topic, and then afterwards we meet again to debrief a little bit and then, again, to see if teachers had any questions, how did that work? And then when we meet the next PLC, then we just kind of reflect and see how we as a group celebrated successes or if we had any barriers or something that we want to look at those*". (control coach)

One coach who was providing "general coaching" (i.e., not linked to a PLC) also based her choice of focus on an initial needs assessment (or later in the year, a previous observation) and a conversation with the teacher.

Coaching processes. During their interviews, coaches were asked to report on the types of coaching strategies they regularly use with their teachers. All coaches reported observing their teachers, sharing resources, and delivering professional development. Two coaches from each condition also discussed classroom observation data and student achievement data with teachers. All CHALK coaches reported modeling lessons, providing observation feedback, and creating goals with their teachers, while only two of the business-as-usual coaches reported using these strategies. However, while two coaches from the business-as-usual condition reported writing lesson plans with teachers, none of the CHALK coaches reported using that strategy.

All coaches interviewed reported having established relationships with the teachers with whom they were working, and they individualized their coaching to those teachers and their classrooms. Five of the seven coaches used structured observation tools (such as fidelity checklists from the PLC curriculum, the categories of the encounter form provided by their program, or even a classroom's CLASS score, when that was available) as a starting point. However, multiple coaches talked about balancing the utility of a checklist with the importance of building relationships with teachers. One coach said

> "*[coaches] walk in the classroom and you have this clipboard or a notebook, they're like, oh. It makes [teachers] a bit distant and maybe uneasy. But then I say, okay. I'll put this notebook away. I will just hang out with you guys*". (control coach)

Building relationships during observations helped coaches better meet teachers' needs. Once they had observed with a mental checklist or topic in mind, coaches moved on to modeling practice (two coaches), whisper coaching (three coaches), interacting with children (three coaches), or checking in with a teacher about their use of a strategy that they had discussed previously and their sense of its efficacy (one coach).

As noted above, all of the coaches used the topic of the PLC to determine at least broadly how to focus their coaching, but two coaches reported that as they became more comfortable in a teacher's classroom, they would start off with the PLC topic and then shift to other areas that they noticed in the classroom during the observation (such as, that the children were all clumped in one area, that children seemed to avoid certain activities because of the way the classroom was organized, or that the teacher could use more positive reinforcement rather than negative redirections). General areas that coaches reported paying attention to were learning in the classroom (what are children doing? How are they interacting?) (two coaches), non-instructional time such as mealtimes or transitions (one coach), and activities and behaviors that could lead to improvement in CLASS scores (two coaches). Most of the coaches were not focused on CLASS scores, but two found

that thinking about interactions and structures that could support improvement in CLASS domains to be a useful heuristic.

### 3.2. Impact Study

Teachers' Classroom Practices. Descriptive statistics for the three observations conducted over the course of the school year are shown below in Table 3. In general, differences between CHALK and coaching-as-usual classrooms across the three observation periods were minimal. Changes over time within the CHALK and coaching-as-usual classrooms were also small. There are no consistent indications that CHALK classrooms exhibited more or less of the targeted classroom practices over the course of the study.

**Table 3.** Descriptive statistics by condition for the practices addressed in the CHALK tool.

| | CHALK | | | | | Control | | | | |
|---|---|---|---|---|---|---|---|---|---|---|
| **Observation 1** | *M* | *N* | *sd* | **Min** | **Max** | *M* | *N* | *sd* | **Min** | **Max** |
| **1: Transitions** | | | | | | | | | | |
| Proportion of sweeps | 0.22 | 20 | 0.08 | 0.09 | 0.35 | 0.27 | 14 | 0.10 | 0.15 | 0.53 |
| Average minutes | 49.60 | 20 | 24.21 | 17.02 | 105.18 | 59.30 | 14 | 26.92 | 21.57 | 116.17 |
| **2: Quality of Instruction** | | | | | | | | | | |
| Instruction (1–4) | 1.59 | 20 | 0.30 | 1.00 | 2.25 | 1.60 | 15 | 0.27 | 1.00 | 2.00 |
| **3: Positive Emotional Climate** | | | | | | | | | | |
| Tone (1–5) | 3.56 | 20 | 0.21 | 3.23 | 3.97 | 3.51 | 15 | 0.37 | 2.88 | 4.06 |
| Behavior approving | 0.05 | 20 | 0.07 | 0.00 | 0.27 | 0.05 | 15 | 0.07 | 0.00 | 0.29 |
| Behavior disapproving | 0.03 | 20 | 0.05 | 0.00 | 0.18 | 0.08 | 15 | 0.08 | 0.00 | 0.30 |
| **4: Teachers Listening to Children (%)** | | | | | | | | | | |
| Teacher listening (total) | 0.07 | 20 | 0.09 | 0.00 | 0.28 | 0.04 | 15 | 0.06 | 0.00 | 0.17 |
| Listening to child | 0.06 | 20 | 0.08 | 0.00 | 0.28 | 0.03 | 15 | 0.05 | 0.00 | 0.16 |
| Children talking (total) | 0.20 | 20 | 0.09 | 0.05 | 0.40 | 0.19 | 14 | 0.09 | 0.06 | 0.39 |
| **5: Sequential Activities (%)** | | | | | | | | | | |
| Non-sequential | 0.12 | 20 | 0.07 | 0.01 | 0.26 | 0.13 | 14 | 0.10 | 0.03 | 0.37 |
| Sequential | 0.19 | 20 | 0.06 | 0.05 | 0.34 | 0.16 | 14 | 0.05 | 0.07 | 0.24 |
| **6: Associative, Cooperative Interactions (%)** | | | | | | | | | | |
| Associative | 0.09 | 20 | 0.05 | 0.01 | 0.18 | 0.08 | 14 | 0.06 | 0.02 | 0.20 |
| Cooperative | 0.01 | 20 | 0.02 | 0.00 | 0.10 | 0.01 | 14 | 0.01 | 0.00 | 0.03 |
| **7: Level of Involvement** | | | | | | | | | | |
| Average involvement overall | 1.91 | 20 | 0.19 | 1.60 | 2.20 | 1.86 | 14 | 0.25 | 1.42 | 2.24 |
| Involvement in learning | 2.68 | 20 | 0.26 | 2.22 | 3.21 | 2.51 | 20 | 0.21 | 2.03 | 2.78 |
| **8: Math Opportunities** | | | | | | | | | | |
| Math focus | 0.04 | 20 | 0.03 | 0.00 | 0.13 | 0.03 | 14 | 0.02 | 0.01 | 0.09 |
| **9: Literacy Opportunities** | | | | | | | | | | |
| Literacy focus | 0.09 | 20 | 0.05 | 0.01 | 0.18 | 0.10 | 14 | 0.06 | 0.00 | 0.21 |
| | **CHALK** | | | | | **Control** | | | | |
| **Observation 2** | *M* | *N* | *sd* | **Min** | **Max** | *M* | *N* | *sd* | **Min** | **Max** |
| **1: Transitions** | | | | | | | | | | |
| Proportion of sweeps | 0.19 | 16 | 0.08 | 0.04 | 0.35 | 0.22 | 13 | 0.07 | 0.12 | 0.35 |
| Average minutes | 55.20 | 16 | 31.25 | 12.62 | 119.87 | 49.57 | 13 | 18.79 | 22.93 | 88.48 |
| **2: Quality of Instruction** | | | | | | | | | | |
| Instruction (1–4) | 1.53 | 16 | 0.28 | 1.00 | 1.88 | 1.62 | 14 | .27 | 1.25 | 2.00 |
| **3: Positive Emotional Climate** | | | | | | | | | | |
| Tone (1–5) | 3.48 | 16 | 0.35 | 2.58 | 3.95 | 3.69 | 14 | 0.23 | 3.22 | 4.10 |
| Behavior approving | 0.06 | 16 | 0.05 | 0.00 | 0.13 | 0.04 | 14 | 0.03 | 0.00 | 0.10 |
| Behavior disapproving | 0.10 | 16 | 0.13 | 0.00 | 0.50 | 0.03 | 14 | 0.05 | 0.00 | 0.18 |

**Table 3.** *Cont.*

| | CHALK | | | | | Control | | | | |
|---|---|---|---|---|---|---|---|---|---|---|
| **Observation 1** | *M* | *N* | *sd* | **Min** | **Max** | *M* | *N* | *sd* | **Min** | **Max** |
| **4: Teachers Listening to Children (%)** | | | | | | | | | | |
| Teacher listening (total) | 0.04 | 16 | 0.06 | 0.00 | 0.19 | 0.08 | 14 | 0.09 | 0.00 | 0.28 |
| Listening to child | 0.02 | 16 | 0.04 | 0.00 | 0.16 | 0.06 | 14 | 0.07 | 0.00 | 0.20 |
| Children talking (total) | 0.20 | 16 | 0.08 | 0.06 | 0.35 | 0.17 | 13 | 0.07 | 0.04 | 0.28 |
| **5: Sequential Activities (%)** | | | | | | | | | | |
| Non-sequential | 0.14 | 16 | 0.07 | 0.01 | 0.28 | 0.18 | 13 | 0.06 | 0.10 | 0.30 |
| Sequential | 0.22 | 16 | 0.09 | 0.08 | 0.41 | 0.01 | 13 | 0.02 | 0.00 | 0.05 |
| **6: Associative, Cooperative Interactions (%)** | | | | | | | | | | |
| Associative | 0.13 | 16 | 0.08 | 0.02 | 0.30 | 0.09 | 13 | 0.04 | 0.01 | 0.18 |
| Cooperative | 0.01 | 16 | 0.01 | 0.00 | 0.03 | 0.04 | 13 | 0.09 | 0.00 | 0.35 |
| **7: Level of Involvement.** | | | | | | | | | | |
| Average involvement overall | 2.01 | 16 | 0.32 | 1.52 | 2.62 | 1.95 | 13 | 0.17 | 1.70 | 2.22 |
| Involvement in learning | 2.74 | 16 | 0.25 | 2.32 | 3.17 | 2.70 | 13 | 0.22 | 2.23 | 3.04 |
| **8: Math Opportunities** | | | | | | | | | | |
| Math focus | 0.05 | 16 | 0.05 | 0.00 | 0.21 | 0.03 | 13 | 0.03 | 0.00 | 0.10 |
| **9: Literacy Opportunities** | | | | | | | | | | |
| Literacy focus | 0.10 | 16 | 0.06 | 0.00 | 0.21 | 0.07 | 13 | 0.05 | 0.02 | 0.17 |
| | **CHALK** | | | | | **Control** | | | | |
| **Observation 3** | *M* | *N* | *sd* | **Min** | **Max** | *M* | *N* | *sd* | **Min** | **Max** |
| **1: Transitions** | | | | | | | | | | |
| Proportion of sweeps | 0.23 | 11 | 0.11 | 0.14 | 0.53 | 0.34 | 11 | 0.10 | 0.18 | 0.51 |
| Average minutes | 44.06 | 11 | 14.94 | 24.52 | 68.33 | 78.86 | 11 | 23.22 | 45.73 | 122.68 |
| **2: Quality of Instruction** | | | | | | | | | | |
| Instruction (1–4) | 1.56 | 11 | 0.16 | 1.25 | 1.80 | 1.65 | 12 | 0.30 | 1.20 | 2.17 |
| **3: Positive Emotional Climate** | | | | | | | | | | |
| Tone (1–5) | 3.41 | 11 | 0.36 | 2.80 | 3.94 | 3.36 | 12 | 0.32 | 2.89 | 3.83 |
| Behavior approving | 0.06 | 11 | 0.07 | 0.00 | 0.20 | 0.03 | 12 | 0.03 | 0.00 | 0.09 |
| Behavior disapproving | 0.09 | 11 | 0.10 | 0.00 | 0.34 | 0.13 | 12 | 0.09 | 0.00 | 0.26 |
| **4: Teachers Listening to Children (%)** | | | | | | | | | | |
| Teacher listening (total) | 0.02 | 11 | 0.04 | 0.00 | 0.11 | 0.03 | 12 | 0.06 | 0.00 | 0.20 |
| Listening to child | 0.02 | 11 | 0.04 | 0.00 | 0.11 | 0.02 | 12 | 0.04 | 0.00 | 0.13 |
| Children talking (total) | 0.16 | 11 | 0.07 | 0.08 | 0.29 | 0.19 | 11 | 0.07 | 0.07 | 0.29 |
| **5: Sequential Activities (%)** | | | | | | | | | | |
| Non-sequential | 0.16 | 11 | 0.09 | 0.02 | 0.28 | 0.12 | 11 | 0.08 | 0.01 | 0.31 |
| Sequential | 0.18 | 11 | 0.07 | 0.05 | 0.32 | 0.19 | 11 | 0.06 | 0.13 | 0.29 |
| **6: Associative, Cooperative Interactions (%)** | | | | | | | | | | |
| Associative | 0.08 | 11 | 0.04 | 0.02 | 0.16 | 0.11 | 11 | 0.07 | 0.02 | 0.21 |
| Cooperative | 0.00 | 11 | 0.00 | 0.00 | 0.00 | 0.01 | 11 | 0.02 | 0.00 | 0.05 |
| **7: Level of Involvement** | | | | | | | | | | |
| Average involvement | 1.94 | 11 | 0.22 | 1.44 | 2.23 | 1.88 | 11 | 0.14 | 1.71 | 2.17 |
| Involvement in learning | 2.57 | 11 | 0.23 | 2.12 | 2.91 | 2.60 | 11 | 0.19 | 2.33 | 3.00 |
| **8: Math Opportunities** | | | | | | | | | | |
| Math focus | 0.05 | 11 | 0.03 | 0.01 | 0.11 | 0.03 | 11 | 0.03 | 0.01 | 0.08 |
| **9: Literacy Opportunities** | | | | | | | | | | |
| Literacy focus | 0.11 | 11 | 0.05 | 0.03 | 0.18 | 0.11 | 11 | 0.07 | 0.02 | 0.21 |

Student Assessments. Baseline descriptive statistics on the child assessments are shown in Table 4. Overall, the sample is not well balanced across CHALK and usual coaching conditions, with the CHALK group having higher average scores across all assessments. The magnitude of the effect sizes suggests that these differences are practically meaningful. The differences between treatment and control for the letter–word, number sense, and quantitative concepts W scores were statistically significant at the 0.05 level, implying an imbalance between the treatment and control sample beyond what we expect to see due to random chance.

**Table 4.** Baseline child assessment data.

| | | Average Score | | | Sample Size | | *g* | *p*-Value |
| | CHALK | Control | Total | CHALK | Control | Total | | |
|---|---|---|---|---|---|---|---|---|
| Picture Vocabulary | | | | | | | | |
| W Score | 458 (15.7) | 453 (11.3) | 456 (14.1) | 94 | 68 | 162 | 0.29 | 0.30 |
| Letter–Word | | | | | | | | |
| W Score | 329 (28.3) | 319 (18.9) | 325 (25.23) | 94 | 68 | 162 | 0.41 *** | 0.01 |
| Number Sense | | | | | | | | |
| W Score | 419 (20.5) | 411 (16.5) | 416 (19.3) | 93 | 68 | 161 | 0.44 * | 0.09 |
| Quantitative Concepts | | | | | | | | |
| W Score | 413 (15.3) | 406 (9.2) | 410 (13.4) | 87 | 67 | 154 | 0.56 *** | <0.01 |
| Head Toes Knees Shoulders | | | | | | | | |
| Total Score | 8.0 (12.7) | 5.3 (9.1) | 6.8 (11.3) | 83 | 64 | 147 | 0.23 | 0.34 |

Notes: Standard deviations in parentheses. Significant difference between groups indicated by *** *p*-value < 0.01, ** *p*-value < 0.05, and * *p*-value < 0.1.

Descriptive statistics on the child outcome assessments and results of the impact analysis are shown in Table 5 (seen at the end of the manuscript). After adjusting for initial assessments scores and other child level covariates, we found no statistically significant difference between the children in CHALK group versus the usual coaching group. The unadjusted mean assessment scores were consistently higher in the CHALK group. The unadjusted mean assessment scores were also higher in the CHALK group at the baseline. The adjusted difference between the CHALK group and the control group was near zero for number sense, quantitative concepts, and picture vocabulary. The adjusted differences for HTKS and the letter-word score show the CHALK group having higher adjusted scores at the posttest, but these differences were not statistically significant.

**Table 5.** Child assessment impact results.

| | CHALK | | | | | Control | | | | | Adjusted Difference | *p*-Value |
| | Fall | | Spring | | N | Fall | | Spring | | N | | |
| Assessment | Mean | SD | Mean | SD | | Mean | SD | Mean | SD | | | |
|---|---|---|---|---|---|---|---|---|---|---|---|---|
| Picture Vocabulary | 459 | 16 | 463 | 15 | 71 | 454 | 12 | 457 | 11 | 48 | −0.62 | 0.83 |
| Letter Word W Score | 331 | 29 | 340 | 27 | 71 | 320 | 20 | 327 | 21 | 48 | 2.29 | 0.57 |
| Number Sense W Score | 421 | 19 | 428 | 20 | 70 | 413 | 18 | 418 | 16 | 48 | −0.14 | 0.96 |
| Quantitative Concepts W Score | 414 | 16 | 419 | 17 | 66 | 408 | 9 | 411 | 13 | 48 | 0.23 | 0.92 |
| Head Toes Knees Shoulders | 16 | 17 | 19 | 18 | 64 | 12 | 13 | 9 | 11 | 47 | 3.56 | 0.29 |

Note. The adjusted difference is measured in points on the assessment scale.

## 4. Discussion

Research indicates that attending high-quality Pre-K has far-reaching effects. More immediately, attending Pre-K prepares students for elementary school [45–47]. However, the positive outcomes associated with preschool cognitive and behavioral skills extend well into adulthood [48]. With this evidence in mind, it is critical that there are programs in place to support educators to implement high-quality evidence-based instructional practices. The most effective approaches to instructional coaching are those that include regular data collection and goal setting that is informed by data [21]. Our study tested the implementation and effectiveness of an app designed to promote rigorous coaching to improve classroom practices and, ultimately, benefit students.

The best practice in coaching is when coaches perform a needs assessment with a teacher by observing and having a discussion with the teacher about areas in which they feel strongly and where they would like support to strengthen their practice or improve an aspect of their classroom. The objectives the coach and teacher set should also be aligned with the goals for continuous improvement that the teacher has set. With the CHALK tool, we expected that coaches would use the app to collect initial observation data and either narrow the focus of their coaching or ensure that CHALK topics/domains were included in the topics of focus for coaching. The results of our examination of coaching implementation revealed that coaches in both conditions chose to focus their coaching based on content they discussed in their professional learning communities, which laid out the set of core topic areas selected at the program level. While CHALK coaches used more specific coaching strategies that are associated with improved instruction (conducting observations, setting goals with teachers, and modeling lessons), the actual focus of their coaching was not narrowed based on the topics embedded in the app.

CHALK provided a structure that may have encouraged coaches to incorporate more of the specific coaching strategies than coaches in the business-as-usual condition. CHALK guides coaches through a process of structured observations, displays results instantly, and has an embedded debrief conversation planning tool and action plan template, all accessible after an observation is completed. In the coaches' training to use CHALK, they were encouraged to follow a coaching cycle that uses these functions—observe, examine results, debrief with teachers, set goals, observe again and monitor progress. Thus, the structure of the app may have had an influence on the strategies coaches chose to use in their work with their teachers.

Regardless, in this study, we did not see a statistically meaningful difference between the classroom practices of teachers in the CHALK condition and those of teachers in the control condition. Thus, we cannot conclude that the specific strategies employed by the CHALK coaches led to greater between-group differences. It is possible that more consistent use of the app or the coaching strategies more commonly seen in the CHALK condition would yield changes in teacher practices over time.

It is important to note that this study took place in a context featuring several forms of support for high-quality teaching and learning. The programmatic and practical aspects of the business-as-usual coaching condition, the context and the counterfactual for our treatment condition, constitute a good environment for implementation. We drew our sample from a high-quality umbrella program that encompasses a range of early care settings including public school preschools, Head Start, and family childcare. The program supports high-quality teaching and learning in several ways. It explicitly espouses high-quality and equitable early learning. It offers professional development for teachers and instructional coaches that is aligned with best practices and tailored to the specific needs of the community. It offers additional workshops on specific topics, outside of the professional learning community format that is standard for the program. Its coaches are supervised by a lead coach, a peer who has experience as an instructional coach in the program; they meet regularly as a group as well as one-on-one, as needed. Finally, it includes a quality improvement and monitoring system. Context is important as coaches in both conditions had resources and ongoing support to provide high-quality coaching to their teachers. Coaches in contexts without these forms of high-level programmatic support may be in a position to benefit from a digital tool such as CHALK, that incorporates some of the coaching practices that have been found to be associated with improving teachers' practices and promoting students' achievement, such as classroom observation [15], coaching conversations that promote teacher reflection [17], goal setting [21] and monitoring progress over time [15].

In addition to our findings from the classroom observations, across all student outcomes assessed in the study, there were no statistically significant differences between the CHALK and control coaching conditions. Students exhibited gains over the school year, especially in terms of the language and literacy outcomes, but the students in the CHALK

condition did not evidence greater (or smaller) gains than their counterparts did in the control group.

It is possible that using CHALK could have produced significant effects if we were to follow the sample longer. This study took place over a single school year. Coaches were trained on the tool in September and used CHALK with varying degrees of consistency and frequency (ranging from weekly to monthly), students were assessed in the fall and spring, and teachers were observed three times over the course of the year. Adopting a new digital tool and looking for immediate positive effects may be unrealistic. Indeed, it is not uncommon to see minimal or null effects of professional development interventions that are evaluated in the first year of implementation [49]. Longer periods of implementation may create a circumstance in which there is greater commitment to implementation, leading to an increased likelihood that positive effects may be found. For example, Crawford et al. [50] developed and tested another digital tool, the Classroom Observation Tool (COT), designed to help inform coaching in Pre-K, and ultimately found positive literacy outcomes for students. However, in their evaluation, the researchers described that COT was adopted over a three-year period, giving coaches and teachers significantly more time to adapt to the COT and participate in ongoing trainings to reinforce their learning and implementation.

Although we did not find significant differences in classroom practices and students assessment gains across the school year, this study yielded valuable information from qualitative data to inform future training, testing, and dissemination of the CHALK tool. For example, we found that coaches implementing CHALK required more ongoing support than we had anticipated. Moving forward, it would be important to provide regular check ins, and additional support for both using the CHALK tool and connecting its content with content specific to their curricula and program requirements. Additional research is needed to gauge the effectiveness of coaching with CHALK as opposed to other coaching frameworks.

*Limitations*

There are important limitations to this study, some that were directly related to the timing of the study, as the COVID-19 pandemic was ongoing. In addition to pandemic-related effects on instruction and on coaching, we found that coaches in both conditions were not consistent in terms of logging their coaching interactions. This led to unreliable data, which limited our ability to compare the groups in terms of aspects of their implementation of coaching, which was an important aim of this study. Thus, we addressed research questions related to coaching implementation using primarily qualitative data from interviews.

Moreover, in interviews with coaches, they indicated that many participants were experiencing considerable distress in coping with the pandemic. For example, one coach said "The teachers are very—they're exhausted, and they're stressed. So sometimes I just listen and it's just a sounding board". (control coach).

Another said, "Everyone's—all the teachers have been all over the place and they really struggled. And I think that's been the biggest piece of just being there and listening... *Hear me* is all anybody really wants. And then they can come and they feel safe and connected. Then we can problem solve". (CHALK coach).

Student and teacher absences were common, and the different preschool centers had periods of closure during the school year. Moreover, coaching was conducted virtually in many cases, which has been found to be less effective than face-to-face coaching is, though is more effective compared to not receiving any modality of coaching [9].

Studies on the effects of the pandemic for early learners suggest that there have been significant negative effects on students' social–emotional development and kindergarten readiness [51]. However, we also know that educators encountered stressors that undoubtedly affected their ability to participate in this research with consistency. Indeed, the results of a qualitative study [52] indicated that teachers increasingly prioritized their students' health and needs above their own, which led to poorer mental health. Researchers also noted the extra burden on teachers in that they were often teaching students and their

parents how to engage in remote learning simultaneously. Moreover, using an ecological perspective, Hanno et al. [53] found that contextual factors (e.g., working in a public school v. family child care) were associated with differential access to resources that might have helped minimize teachers' pandemic-related stress. Thus, coaches' shift from prioritizing a focus on improving specific teacher practices at a time when teachers were understandably in need of social support may have been the most appropriate response at the time. In support of this, recent research found that educators that received more social–emotional support during the COVID-19 pandemic reported experiencing lower levels of challenge with implementing distance learning, and had lower burnout and self-judgement [54]—all critical to teachers' ability to support students as they struggled with their own challenges.

Another limitation of our study was the sample size. Our small sample size meant we were under-powered. In addition to this, despite the fact that our sample was racially diverse, the small sample size, all from a single school district, limits the generalizability of our findings. This, coupled with the short study duration, limited the ability to detect findings, positive or negative, that may have been associated with use of the CHALK app. It is important to pay attention to null findings in well-powered evaluation studies [55], as those have important implications for the field, but null findings from the current study point to the need for additional research investigating the efficacy of such a tool.

**Supplementary Materials:** The following supporting information can be downloaded at: www. chalkcoaching.com; coaching app demonstration video.

**Author Contributions:** Conceptualization, C.C. and S.J.W.; methodology, C.C., S.J.W. and M.W.F.; training quantitative data collectors, M.W.F.; management of quantitative data collection, M.W.F.; data cleaning, M.W.F.; formal analysis of quantitative data, S.L.; formal analysis of qualitative data, C.L.; data curation, C.L. for qualitative data and S.L. for quantitative data.; writing—original draft preparation, S.J.W., C.L. and C.C.; writing—review and editing, C.C.; supervision, C.C. and S.J.W.; project administration, S.J.W. and C.C.; funding acquisition, C.C. All authors have read and agreed to the published version of the manuscript.

**Funding:** This research was funded by the National Science Foundation, grant number DRK-12 18138008.

**Institutional Review Board Statement:** The study was conducted in accordance with the Declaration of Helsinki, and approved by the Institutional Review Board of the University of Dayton, approved 5 May 2020 (Approval #20799481). Data were shared with Abt Associates for analysis after both entities signed a Data Use Agreement, executed on 7 April 2022.

**Informed Consent Statement:** Informed consent was obtained from all subjects involved in the study.

**Data Availability Statement:** The data presented in this study are available on request from the corresponding author. The data are not publicly available due to the small sample size and risk of re-identification of confidential data.

**Acknowledgments:** The author gratefully acknowledges the instructional coaches, teachers, administrators, and software developers that were part of the iterative design and development phase of this work in addition to the teachers and coaches that participated in the evaluation study. Finally, Sandra Wilson and Mary Fuhs, that led the Abt Associates and University of Dayton teams, respectively, were vital to this work.

**Conflicts of Interest:** It should be acknowledged that the author was the lead investigator on the research team that developed the CHALK tool. Results of the evaluation study comparing this tool coaching-as-usual, while null, are presented here for transparency.

## Abbreviation

| Term | Abbreviation |
| --- | --- |
| Prekindergarten | Pre-K |
| Early Childhood Environmental Rating Scale | ECERS |
| Classroom Assessment Scoring System | CLASS |
| Coaching to Help Activate Learning for Kids | CHALK |
| Child Development Associate | CDA |

| Term | Abbreviation |
|------|--------------|
| Professional learning communities | PLC |
| Teacher Observation in Preschool | TOP |
| Child Observation in Preschool | COP |
| Head Toes Knees Shoulders | HTKS |

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
