# Peer review of "Preliminary Evaluation of a Mobile, Web-Based Coaching Tool to Improve Pre-K Classroom Practices and Enhance Learning"

_education, doi:10.3390/educsci13060542_

Round 1

Reviewer 1 Report

I appreciate the opportunity to read this essay, so thank you very much. 

The paper's title is relevant to this research.

The statistical analysis supports the theme, which is reasonably current and significant.

The goal is apparent, but the author must ensure readers understand how their research will affect students' future education.

The authors should analyze the variables influencing students' evolutionary development. Possibly addressed? What needs to be changed 

The writers made an effort to explain the data. There is no repetition in the text in the results. The findings are well explained. The authors contextualized interesting viewpoints and multiple perspectives without overinterpreting them. 

Generally, references or outcomes are used to substantiate the conclusions. The article might benefit from an improved list of citations from the last 3 years.

Mainly the study's design was suitable for achieving the goal. This article has no significant errors in my opinion, and it is coherent throughout.

The main weak point of the article is the sample and the very short time of the experiment. The article might represent preliminary research, that has to be reloaded. I would recommend publishing the article after testing CHALK Tool for a longer period to ensure that the results are representative and can be extrapolated over the entire population.

I am not sure about publishing this article. The results are not very representative. The study needs a longer period to be sure about conclusions.

Author Response

Reviewer 1  
Comment Author Response
I appreciate the opportunity to read this essay, so thank you very much.  The paper's title is relevant to this research. The statistical analysis supports the theme, which is reasonably current and significant.  Thank you for your review!  We appreciate your thorough reading and suggestions to improve our manuscript.
The goal is apparent, but the author must ensure readers understand how their research will affect students' future education. The authors should analyze the variables influencing students' evolutionary development. Possibly addressed? What needs to be changed. We appreciate this feedback as it is important to highlight the long-term implications of students’ early learning experiences. To ensure we adequately address the implications, we have added sections to both the introduction and the discussion, highlighting the importance of promoting quality classroom practices in Pre-K as students’ early learning experiences are predictive of a myriad of outcomes including high school graduation and better health in adulthood.  In the introduction (in the revised manuscript, lines 27-40, the text now reads, “Children’s early learning experiences have long-term effects on their academic and social-emotional development.  Research investigating these effects has found that attending high quality prekindergarten (Pre-K) programs is associated with less grade retention and increased high school graduation rates (Barnett & Jung, 2021), and greater earnings, better health, and less involvement in crime in adulthood (Reynolds et al., 2011). Importantly, these benefits are even more pronounced for students from economically disadvantaged backgrounds (Blau, 2021), suggesting that early learning experiences may help to close the opportunity gap that exists between lower and higher socioeconomic students. Moreover, cost-benefit analyses of high-quality Pre-K points to substantial economic benefits for society at large (e.g., Elango et al, 2016; Heckman et al., 2010). For these reasons, it is critical that educators provide high quality learning opportunities in Pre-K, most children’s first exposure to an organized learning environment, which can be built upon in subsequent grades. To accomplish this, educators need ongoing support as they are relied upon to set students on a positive trajectory.

In the discussion, the following sentences were added, including citations from research published within the past 3 years: “Research indicates that attending high quality Pre-K has far-reaching effects.  More immediately, attending Pre-K prepares students for elementary school (Ansari et al., 2021; Sulik et al., 2023). But the positive outcomes associated with preschool cognitive and behavioral skills extend well into adulthood (Pan et al., 2023).  With this evidence in mind, it is critical that there are programs in place to support educators to implement high quality evidence-based instructional practices.”
The writers made an effort to explain the data. There is no repetition in the text in the results. The findings are well explained. The authors contextualized interesting viewpoints and multiple perspectives without overinterpreting them. Generally, references or outcomes are used to substantiate the conclusions. The article might benefit from an improved list of citations from the last 3 years. Thank you for your feedback. We have now included more recent references to substantiate our conclusions. For example, in section 1.3. Leveraging Technology to Support Professional Development, we now include a 2023 study by Ho and colleagues, which describes research on another web-based professional development program. In addition, new sections added to the introduction and discussion include recent research.  In addition to portions of those sections describing long-term effects of attending high quality Pre-K, we bring in COVID-19 related research from 2021 that describes circumstances under which teachers benefit from social-emotional support (lines 660-667).
Mainly the study's design was suitable for achieving the goal. This article has no significant errors in my opinion, and it is coherent throughout. The main weak point of the article is the sample and the very short time of the experiment. The article might represent preliminary research, that has to be reloaded. I would recommend publishing the article after testing CHALK Tool for a longer period to ensure that the results are representative and can be extrapolated over the entire population. While we do include statements acknowledging our small sample size as a limitation (see lines 668-669 and 672-675), we have added a statement about how a combination of a small sample size and shorter study duration both likely contribute to minimizing the chance of detecting findings (lines 669-672). We also added a statement regarding the lack of generalizability, “In addition to this, despite the fact that our sample was racially diverse, a small sample size, all from a single school district, limits the generalizability of our findings.”

In terms of the short time of the experiment, we added the following (lines 608-612): “Indeed, it is not uncommon to see minimal or null effects of professional development interventions that are evaluated in the first year of implementation (e.g., U.S. Department of Education, 2008).  Longer periods of implementation may create a circumstance in which there is greater fidelity of implementation, leading to an increased likelihood that positive effects may be found.”
I am not sure about publishing this article. The results are not very representative. The study needs a longer period to be sure about conclusions. We acknowledge this as a valid concern. Through making the aforementioned changes and additions to the text, we hope we have addressed this concern.  

Reviewer 2 Report

In this time of limited professional development time and funding and increased availability of technology, it is time that the teaching and learning community look to effective ways to enhance teaching through the use of mobile-web-based coaching tools, such as the one described in this article.

While the research did not reveal significant differences in classroom practices and students' assessment gains, it did produce qualitative data that could support future research in this area. For this reason, it is worthy of publication.

Author Response

Reviewer 2  
Comment Author Response
In this time of limited professional development time and funding and increased availability of technology, it is time that the teaching and learning community look to effective ways to enhance teaching through the use of mobile-web-based coaching tools, such as the one described in this article. While the research did not reveal significant differences in classroom practices and students' assessment gains, it did produce qualitative data that could support future research in this area. For this reason, it is worthy of publication. Thank you for your review!  We hope that our work makes a valuable contribution and are gratified that you feel it is worthy of publication.

Reviewer 3 Report

The paper titled " Preliminary Evaluation of a Mobile, Web-based Coaching Tool to Improve Pre-K Classroom Practices and Enhance Learning" is reviewed. A number of related recommendations are presented as follows:

·       The author/(s) should avoid words expressing ownership. For example: don't use “our work” or “we used”.

·       The authors should report the main of the manuscript in the abstract.

·       In addition, The authors might use an abbreviation table at the beginning or end of the text since lots of abbreviations are used in the text.

·       I feel method section could be expanded to include to describe tests undertaken.

·       It's nice that the Methods section is separated by student and teacher sizes. However, the method along with the application should be specified in detail and regularly. For example, how and according to what students and teachers are selected, how the application is made.

·       It will be more meaningful for the reader to reduce the subheadings and bring them together in a more regular and sequential manner.

·       The limitations may be mentioned in the method section instead of the discussion section.

·       The results section can be conveyed in a more concise and organized manner.

Author Response

Thank you for your review of our manuscript!  Your feedback is much appreciated.

Reviewer 3  
Comment Author Response
The author/(s) should avoid words expressing ownership. For example: don't use “our work” or “we used”. APA style encourages authors to use active voice to "create direct, clear, and concise statements" (https://apastyle.apa.org/style-grammar-guidelines/grammar/active-passive-voice), which the reviewer may interpret as expressing ownership. However, we are reluctant to change to using passive voice, as it could lead to sentences that are less clear to most readers.
In addition, The authors might use an abbreviation table at the beginning or end of the text since lots of abbreviations are used in the text. Thank you for this suggestion. We have now added an abbreviations table to the end of the text.
I feel method section could be expanded to include to describe tests undertaken While we appreciate the suggestion to expand the assessment descriptions, we feel that they are fairly detailed and are not sure if adding more information would be of value to the readership. We did include a statement indicating that assessments were administered by trained assessors. 
It's nice that the Methods section is separated by student and teacher sizes. However, the method along with the application should be specified in detail and regularly. For example, how and according to what students and teachers are selected, how the application is made. We have now added more information about how CHALK was developed (lines 205-207).  On p. 7 (lines 270-278), we describe how participants were selected.
The limitations may be mentioned in the method section instead of the discussion section. Because a description of the study limitations put our findings in the larger context, we determined it would be most helpful to have them in the discussion section. 
The results section can be conveyed in a more concise and organized manner. We are curious if the reviewer could provide additional guidance on how to make the results section more concise. For example, would it be better to have fewer tables and report more findings in the text?  Would the reviewer suggest that we omit some participants' quotes from qualitative data collection?

Reviewer 4 Report

I commend you on your work in featuring an area of education often overlooked. Despite the fact that little difference was detected it seems like digital helpers and tools are well received in this setting and the need for continuous review and development of teaching skills established. Seeing much of the same conclusions in higher secondary education I wonder about the implementation as well and enjoyed to learn your ideas and thoughts about it.
I also very much agree on the negative mental health effects of COVID in the teaching profession and appreciate you adding it to the limitations of this study.

I am looking forward to your published paper and your further work in this interesting area.

Author Response

Reviewer 4  
Comment Response
I commend you on your work in featuring an area of education often overlooked. Despite the fact that little difference was detected it seems like digital helpers and tools are well received in this setting and the need for continuous review and development of teaching skills established. Seeing much of the same conclusions in higher secondary education I wonder about the implementation as well and enjoyed to learn your ideas and thoughts about it. Thank you for your review!  We very much appreciate your feedback on our manuscript.
I also very much agree on the negative mental health effects of COVID in the teaching profession and appreciate you adding it to the limitations of this study. We appreciate the reviewers’ comment.  With the acknowledgement of the negative effects of COVID on the teaching profession, we added further support from research about the appropriateness of coaches’ strategies for supporting teachers in our study, stating, “Thus, coaches’ shift from prioritizing a focus on improving specific teacher practices at a time when teachers were understandably in need of social support may have been the most appropriate response at the time.  In support of this, recent research found that educators that received more social-emotional support during the COVID-19 pandemic reported experiencing lower levels of challenge with implementing distance learning, and had lower burnout and self-judgement (Zieher et al., 2021) – all critical to teachers’ ability to support students as they struggled with their own challenges.”

Round 2

Reviewer 1 Report

Dear author 

The paragraph inserted in the introduction must be introduced in the Discussion or Conclusion.

I do not consider that the authors answered all my concerns, especially regarding references.

Author Response

Response to Reviewer 1 Comments on Revised Manuscript

We appreciate the reviewer’s additional feedback on our revised manuscript!  Below we describe how we have addressed each concern.

Point 1: The paragraph inserted in the introduction must be introduced in the Discussion or Conclusion. 

Response 1: We agree it is important to revisit the information we included in the introduction.  The beginning of the Discussion includes the following sentences that highlight the importance of this research with implications for children’s development into adulthood: “Research indicates that attending high quality Pre-K has far-reaching effects.  More immediately, attending Pre-K prepares students for elementary school [45-47]. But the positive outcomes associated with preschool cognitive and behavioral skills extend well into adulthood [48].  With this evidence in mind, it is critical that there are programs in place to support educators to implement high quality evidence-based instructional practices. The most effective approaches to instructional coaching are those that include regular data collection and goal setting that is informed by data [21].  Our study tested the implementation and effectiveness of an app designed to promote rigorous coaching to improve classroom practices and, ultimately, benefit students.

Point 2: I do not consider that the authors answered all my concerns, especially regarding references.

Response 2: We do not read the reviewer’s comment about references to suggest that we should replace older studies with newer ones; rather, we added some recent work in support of our overall argument/approach.  We have now included additional recently published research relevant to our study. With this round of additional citations, 43% of the references we include in the manuscript are drawn from 2020-2023.